# Why Is Longevity Still a Scientific Mystery? Sirtuins—Past, Present and Future

**DOI:** 10.3390/ijms24010728

**Published:** 2022-12-31

**Authors:** Patrycja Ziętara, Marta Dziewięcka, Maria Augustyniak

**Affiliations:** Faculty of Natural Sciences, Institute of Biology, Biotechnology and Environmental Protection, University of Silesia in Katowice, ul. Bankowa 9, 40-007 Katowice, Poland

**Keywords:** sirtuins, longevity, modulators

## Abstract

The sirtuin system consists of seven highly conserved regulatory enzymes responsible for metabolism, antioxidant protection, and cell cycle regulation. The great interest in sirtuins is associated with the potential impact on life extension. This article summarizes the latest research on the activity of sirtuins and their role in the aging process. The effects of compounds that modulate the activity of sirtuins were discussed, and in numerous studies, their effectiveness was demonstrated. Attention was paid to the role of a caloric restriction and the risks associated with the influence of careless sirtuin modulation on the organism. It has been shown that low modulators’ bioavailability/retention time is a crucial problem for optimal regulation of the studied pathways. Therefore, a detailed understanding of the modulator structure and potential reactivity with sirtuins in silico studies should precede in vitro and in vivo experiments. The latest achievements in nanobiotechnology make it possible to create promising molecules, but many of them remain in the sphere of plans and concepts. It seems that solving the mystery of longevity will have to wait for new scientific discoveries.

## 1. Introduction

Sirtuins are a group of proteins regulating many physiological functions and life processes. They have become an essential factor in research focused on aging processes as the deletion in the Sir-2 gene, which is responsible for the expression of sirtuins, results in a reduced lifespan of *Saccharomyces cerevisiae* [1]. With high probability, they may be one of the many missing elements on the way to longevity, significantly since the activity of individual sirtuins in cells characteristically decreases with age [2,3]. Geneticist Amar Klar first described sirtuins in the 1970s by discovering the Sir-2 (Yeast Silent Information Regulators II, SIR2) gene in *Saccharomyces cerevisiae* cells. Still, re-search indicating their potential impact on longevity was carried out about 20 years later, in 1991, by Leonard P. Guarante [4,5]. In 1999, by adding SIR2 to the DNA, it was possible to extend the replication life of yeast compared to the wild line. It was also confirmed two years later, extending the life cycle of worms. In 2003, David Sinclair showed that the activity of sirtuins could be modified by other substances, for example, resveratrol. A year later, they showed the connection of SIRT1 with FOXO proteins and proved their neuroprotective effect [6,7,8]. In the following years, there was further re-search on sirtuins, compounds modulating their activity, bioavailability, and their connection with metabolic functions (Figure 1). The focus was on classifying them and analyzing the mechanisms of their impact on various organisms, which, according to many scientists, may be crucial in determining them as regulators of aging [9,10,11].

## 2. General Characteristics of Sirtuins: Classification, Structure, and Localization

Sirtuins are NAD+ cofactor-dependent histone deacetylases (class III–HDAC). Seven mammalian sirtuins have been identified so far (SIRT1 to SIRT7) [4,12]. The catalytic domain of the NAD+ sirtuins contains 258 amino acid residues on average (the shortest is attributed to SIRT6 of 239 amino acid residues, while the longest is to SIRT2–275 amino acid residues composing) [13,14]. They occur in both bacteria and complex organisms [15] and take part in many processes regulating biological functions (cell cycle, cellular metabolism, genome homeostasis) [4,16,17,18]. It is considered that sirtuins can prevent DNA damage, the formation of cancer cells, and inflammatory reactions. They play a crucial role in the antioxidant defense [19,20]. In the human body, sirtuins are present in many different organs. From the level of the liver and kidneys, they can be involved in reducing stress and inflammatory reactions [21,22] because they have a unique role in the production of cytokines [18]. Assigning individual types of sirtuins to organisms is difficult since data on their presence in specific species are unclear or incomplete and constantly updated. Bioinformatics analysis and spatial visualization of the sirtuin particle show that a sirtuin’s N- and C-terminal regions may be unique within a species. Thus, the catalytic activity and dynamics may differ in unrelated species [23,24]. The current classification of sirtuins depends on the sirtuin location in the cell and is still updated through new research. Three main localization sites have been determined: cytoplasm, mitochondrion, and cell nuclei (Figure 2) [18].

Nuclear sirtuins are histone-modifying deacetylases and are responsible for gene expression. These include SIRT1, SIRT6, and SIRT7, but there are reports of SIRT2 and SIRT3 that can migrate between organelles. Their mission is to determine in the cell when aging will begin [18,25], and probably have cytoprotective activity [26]. In turn, mitochondrial sirtuins are also called imported mitochondrial sirtuins (mtSIRT) with an N-terminal sequence [27]. Mitochondrial sirtuins include: SIRT3, SIRT4 and SIRT5. Regulation of their function occurs in post-translational modification of substrate proteins through deacetylation, demalonylation, lipoamidation, and poly ADP-ribosylation [12]. They can modify and regulate oxidative stress in cells and reduce reactive oxygen species directly in the organism [28,29]. Mitochondrial sirtuins reduced expression may contribute to the disturbance of the body’s homeostasis, ultimately leading to the formation of pathologies and diseases related to the aging process, mainly affecting organs such as skeletal muscles, liver, pancreas, heart, or bones [25,30,31]. Cytoplasmic sirtuins coordinate the processes associated with cell apoptosis and DNA repair [32]. Sirtuin 2, deacetylating α-tubulin, is mainly located in the cytosol [33]. Sirtuin 1, on the other hand, is most found in the cell nucleus but can migrate and be present in the cytoplasm [34,35]. It regulates the activity of proteins responsible for chromatin structure, holds the cell cycle, and maintains homeostasis [36]. Explicit determination and understanding of the biological role of sirtuins are difficult due to their function in many biochemical pathways and the lack of inhibitors directed against specific types of sirtuins. Current studies are based on presumptions, defining newly discovered compounds as potential or promising activators [37,38].

### 2.1. Sirtuin 1 (SIRT1)

Sirtuin 1 is the human homolog to Sir2, derived from *Saccharomyces cerevisiae* (yeast) [39]. It is localized in the cell nucleus, but numerous reports in the literature about its occurrence in the cytoplasm [4]. It is the best-tested and well-known sirtuin so far [40]. SIRT1 affects insulin secretion from pancreatic beta cells, oxidation, fatty acid synthase induction, transcriptional activity, and deacetylation of histones and non-histone proteins [41,42,43]. It may inhibit inflammation, produce pro-inflammatory cytokines, have anti-apoptotic effects, and increase resistance to cellular stress. Moreover, it inhibits the p53 protein [44,45] and helps to maintain organelles integrity (peroxisomes and mitochondria) [46].

### 2.2. Sirtuin 2 (SIRT2)

Sirtuin 2 plays an essential role in oxidative stress protection, reducing its intensity through the deacetylation of FOXO and Nrf2 transcription factor [17,47]. It interacts with the protein p53, histone H4 lysine 16, and α-tubulin [48] and acts as a mitotic checkpoint protein that prevents the formation of hyperploid cells in early metaphase [49]. SIRT2 is also an epigenetic regulator because its dysregulation affects tumor progression (its expression is reduced in reproductive, urinary, or digestive systems cancer) [50,51]. Park et al. (2021) showed that SIRT2 has functions in lipid metabolism in mice, activating hepatocytes and hepatic stellate cells [52]. Furthermore, it is associated with cell proliferation and neurological disorders [53]. Jeong and Cho (2017) state that SIRT2 can regulate neuronal differentiation through the deacetylation of α-tubulin but the mechanisms of this process are not clarified [54].

### 2.3. Sirtuin 3 (SIRT3)

The primary location of sirtuin 3 is the mitochondria. However, it can migrate to the cell nucleus or cytoplasm [55]. It regulates lysine acetylation in mitochondria [56]. In humans, it affects the stabilization of heterochromatin and cytosol [25,57]. Some studies indicate that it appears in the mitochondrion only due to cellular stress, moving there from the cell nucleus [58]. Sirtuin 3 activates the mitochondrial genes PGC-1a and UCP-1, which suggests its essential role in thermogenesis [59]. It is associated with the longevity gene in men [60] and may prevent bone loss in mice by regulating E11/gp38 [61]. According to Liu et al. (2022), SIRT3 activation might reduce the anti-inflammatory effect by weakening NF-κB signaling [62]. SIRT3 is a kind of mitochondrial protector that can prevent cardiovascular diseases because it participates in their metabolism and dynamics by regulating the acetylation of mitochondrial proteins [63]. Its connection to cardiovascular diseases was also demonstrated by Liu et al. (2021) in studies on the development and regression of atherosclerosis. SIRT3 participating in acyl modifications maintains mitochondrial homeostasis and prevents vasculitis. However, there is scarce information about its use as a therapeutic against atherosclerotic lesions, which suggests the need for further research [64].

### 2.4. Sirtuin 4 (SIRT4)

Jaiswal et al. (2022) reported that SIRT4 is localized only in the mitochondrial matrix [57]. It is known that due to its location, it does not participate directly in DNA repair, but it shows strong enzymatic activity against oxidative stress generated during cellular respiration. It can protect the genome indirectly by lowering glutamate dehydrogenase activity and arresting the cell cycle. It participates in mitochondrial ATP homeostasis and, with other sirtuins, in stem cell differentiation [57]. Aside from its mitochondrial role, further enzymatic activity is puzzling.

Studies indicate its function in the oxidation of fatty acids in the muscles and liver and regulating insulin secretion. It probably also plays the role of a mitochondrial tumor suppressor [65]. SIRT4 is little known and described in vertebrates [66,67]. He et al. (2022) suggested that it may inhibit doxorubicin (DOX)-induced cardiotoxicity by regulating the AKT/mTOR/autophagy pathway [68] or prevent the release of pro-inflammatory cytokines IL-1β, TNF-α and IL-6 [69].

### 2.5. Sirtuin 5 (SIRT5)

SIRT5 is classified as mitochondrial and regulates metabolic pathways such as the Krebs and fatty acid cycles. Probably SIRT5 polymorphisms result in shortened human lifespan [28]. Sirtuin 5 affects brown adipose tissue activity as it can regulate mitochondrial respiration and protein kinase activity after treatment with the sirtuin inhibitor MC3482 [70]. It can catalyze deglutarylation, demalonylation, and desuccinylation reactions while exhibiting low deacetylation activity [37,71]. Jung et al. (2022) showed the influence of SIRT5 on mitochondrial respiration, and the TNF-α factor induces its expression, contributing to the delaying of the aging process [72].

### 2.6. Sirtuin 6 (SIRT6)

Sirtuin 6 has broad molecular functions and is a critical factor in the aging process due to its increased activity in the cell [73]. It is most often found in the cell nucleus. However, there are many reports of its occurrence in the cytosol [25]. It is related to glucose and lipid metabolism in the liver and pancreas via FoxO1 [74]. Research conducted on SIRT6 a few years earlier indicated its genetic instability, and some studies even proved that it caused premature aging processes. SIRT62/2 mice die prematurely and exhibit severe defects such as lymphopenia, loss of subcutaneous fat, reduced bone mineral density, and impaired glucose homeostasis. These are similar pathologies to those occurring in seniors [75]. Then, it was shown how its overexpression affects the extension of the life of mice [3]. According to Carreno et al. (2020), age-related diseases may lead to oxidative DNA damage. Sirtuin 6 may prevent these processes while stabilizing the genome and serving as a diagnostic biomarker [76,77]. SIRT6 has become one of the leading research subjects, thus diverting attention from SIRT 1 due to its more significant therapeutic potential, especially in neurodegenerative diseases [2].

### 2.7. Sirtuin 7 (SIRT7)

It is the only sirtuin located in the nucleolus, is involved in stress response and ribosome biogenesis, and is the least described among the sirtuins [78]. Sirtuin 7 interacts with RNA polymerase (Pol I) and histones to regulate rDNA transcription (transcript elongation). Its inhibition induces a decrease in transcription, stops cell proliferation, and initiates apoptosis [79]. Presumably, it may also inhibit cell apoptosis under stress conditions (e.g., hypoxia) [78]. According to Li et al. (2022), sirtuin 7 may inhibit the NF-κB signaling pathway, activated after entering an external factor into the body, resulting in the inhibition of inflammation caused by inflammation lipopolysaccharides [80]. SIRT7 activity is associated with the pathology of the cardiovascular system in the form of heart failure, atherosclerotic changes, and myocardial hypertrophy, thus is considered a potential therapeutic target against cardiovascular and kidney diseases [81,82]. There have also been reports that it may affect mitochondrial activity [83].

### 2.8. Sirtuin 8—Myth or Fact?

There are few reports on the discovery of SIRT8 expression. Sun et al. reported its activity in thyroid cancer, which was later contradicted by classifying this sirtuin as SIRT7 [84]. Other studies on the *Locusta migratoria* genome showed that this species has a higher amount of enzymes affecting the modification of histones compared to other insects and indicated the presence of a gene similar to SIRT3 in mammals (regulation of mitochondrial fatty acid oxidation). It was proposed to create a new subfamily for SIR2-Sirt8, but from the experiment until 2022, no new classification was recorded due to the lack of subsequent independent analyses [85].

### 2.9. Sirtuins and Invertebrates

Sirtuins are mainly described in vertebrates, where mice (*Mus musculus*) are a significant research object. So far, it has not been possible to identify sirtuins in invertebrates fully, and the information, sometimes contradictory, relates only to selected species. It was found that insect sirtuins are significantly different from mammalian ones. Research conducted on *Caenorhabditis elegans* confirmed the presence of the SIR2 (Sir-2.1) ortholog, which increases the viability of individuals by almost 50%. It depends on the transcription factor Daf-16 and is regulated in the insulin/IGF-1 pathway [86]. Other results concern *Drosophila melanogaster.* It has been described that the dSir2 gene does not play a significant role in the viability of individuals but is only a regulator of heterochromatin formation. Individuals with a mutation in the dSir-2 gene are not characterized by a shorter life cycle like it is in worms [87]. Shukla et al. (2022) proved that the SIRT6 protein in *D. melanogaster* can potentially slow down larval development [88]. Other studies on this species have shown that SIRT4 has a regulatory effect on oxidative metabolism and may promote longevity. Overexpression of SIRT2 in muscles and neurons enables restoration of exercise capacity in *Drosophila* with mitochondrial dysfunction [89,90,91,92]. Wang et al. (2021) identified in *Ruditapes philippinarum* five types of sirtuins: SIRT1, SIRT2, SIRT4, SIRT6, and SIRT7, the expression of which depends on the developmental stage of the individual and environmental conditions. They play a role in restoring homeostasis after exposure of individuals to air [15]. Zhang et al. (2022) showed that in *Bombyx mori* (silkworm), SIRT2 and SIRT5 could enhance antiviral immunity, and the SIRT5 inhibitor (suramin) promotes nucleopolyhedrovirus replication [93]. Three years earlier, in 2019, Brent et al. reported inhibition of arbovirus genetic replication in adult flies treated with SIRT1 and SIRT2 inhibitors, which may contribute to a more efficient fight against diseases transmitted by forceps and mosquitoes (yellow fever, dengue fever, West Nile fever) [94]. In turn, May et al. (2021) presume that sirtuins play a role in regulating heat shock in *Mytilus californianus* (Conrad). However, determining the role of specific sirtuins in regulating this process requires more precise and targeted research [95].

## 3. Enzymatic Activity

Due to the complicated mechanism and many correlations, the enzymatic functions of sirtuins have not yet been summarized. Existing studies on sirtuin activity across species are inconsistent. The mechanism of stimulation and inhibition of specific sirtuins is still not fully understood, which makes it challenging to create the activators that must be matched to a particular sirtuin/substrate pair. It is known that sirtuins, as class III HDACs require the NAD+ cofactor to catalyze the deacetylation of substrate proteins. The reaction proceeds to produce a deacetylated protein product in the form of nicotinamide and O-acetyl-ADP-ribose, which is a functional difference compared to class I and class II HDACs [96].

Modulating repair pathways for which sirtuins are responsible occurs through the deacetylation of repair factors, which changes the availability and cohesion of chromatin [58]. For example, SIRT1 directly affects histone deacetylation by structural remodeling chromatin and silencing the transcription process [58]. According to Sun et al. (2022), inhibition of sirtuin 1 may additionally regulate the deacetylation reaction in mitochondria, although many targets of SIRT1 action have primarily been identified as nuclear [97]. Sirtuins serve as a catalyst in lysine deacetylation reactions that require NAD+ as a co-substrate [98]. Sirtuins, as NAD+-dependent deacetylases, can remove various acyl groups from acylated lysines in histones and non-histone proteins. They are involved in the activity of transferases. The catalytic mechanisms of sirtuins are not fully known and described. However, studies indicate their catalytic potential in the case of ADP-ribose transfer to the acceptor medium [99]. Moreover, sirtuins are activated during malnutrition. In a caloric restriction/starvation, all sirtuin isoforms are activated (Figure 3) due to the increased activity of the electron transport chain (respiratory chain), which leads to an increase in the ratio of NAD+ to NADH [100].

### 3.1. The Sirtuin Activity in the Light of Exercise Training and Caloric Restriction

Proper metabolism and homeostasis, various environmental factors, physical activity, and nutrients of consumed food are only selected components affecting the organism’s condition. Both nutrition and exercise training are thought to delay the aging process in many species indirectly. Studies in rats have shown that swimming training inhibits inflammatory signaling and Fas-dependent apoptotic pathways in the hippocampus of the aging brain. In the group of physically active animals, signaling associated with longevity (AMPK/SIRT1/PGC-1α) increased significantly, as did the IGF1/PI3K/Akt pathway (brain survival) [101]. Sellitto et al. (2022) recognized exercise training as a natural activator of SIRT1 in humans. A higher level of SIRT1 mRNA characterized middle-distance runners compared to people leading a sedentary lifestyle. Interestingly, taking dietary supplements such as vitamin C, vitamin E, and mineral salts during training hindered the activation of SIRT1 by physical exercise [102]. Physical activity stimulates the expression of SIRT1 and SIRT3 in skeletal muscles by regulating oxidative metabolism (MnSOD, FOXO3a), mitochondrial biogenesis, and enhancing ATP production [103]. Edget et al. (2016) explained that single workouts do not cause genetic expression of SIRT3, to increase the amount of protein, effort must be a repeated stimulus [104]. In addition, the expression of sirtuins in skeletal muscle depends on the type of exercise. Increased AMPK phosphorylation via high-intensity interval training activates SIRT1 increasing PGC-1α. In turn, regular aerobic exercise in people leading a sedentary lifestyle stimulates SIRT3, increases the activity of COX, β-hydroxyacyl-CoA dehydrogenase and contributes to the increase of NAMPT, allowing the synthesis of the sirtuin substrate–NAD+ [105,106].

The first information about the caloric restriction and its positive impact on an organism date back to the 20th century. Crucial was caloric restriction without malnutrition [107]. Maldonado et al. (2021) described calorie restriction as the most effective method of delaying the onset of many age-related diseases. Short-term caloric restriction after switching from standard nutrition in mice contributes to increased expression of all seven sirtuin proteins [108]. Furthermore, caloric restriction reduces telomere erosion in various mouse tissues (leukocytes, lungs, kidneys, muscles) observed with overexpression of telomerase reverse transcriptase (TERT) [109]. It has been established that increasing the activity of SIRT1 in combination with other sirtuins stabilizes telomeres and alleviates telomere-related disorders [110]. Increased expression of SIRT1 was also observed in mice switched to a 20% reduced calorie diet after an initial overfeeding period. Caloric restriction reduced hepatic steatosis, decreased superoxide anion levels, and increased catalase and superoxide dismutase protein expression [111]. In addition, it was proved that the energy supply at the level of 70% of the demand increased the expression of SIRT3 in the liver, muscles, and adipose tissue of obese rats [112]. Being overweight significantly contributes to the formation of acute inflammation and oxidative stress in the organism [113]. SIRT2 is essential for the protective effect of caloric restriction against high-fat diet-induced cancer [114]. SIRT1 and SIRT2 expression also increase after reducing the proportion of simple sugars in the diet. Their expression is reduced in type 1 and 2 diabetes [115]. In turn, a high-fructose diet causes high levels of glycation products that inhibit SIRT1 expression and impair muscle performance in mice [116]. A caloric restriction also reduces the adverse effects of excessively accumulated fatty tissue. Opstad et al. (2022) confirmed a strong correlation between sirtuins, obesity, and caloric restriction. They researched obese patients who underwent bariatric surgery. The gastric bypass procedure resulted in a significant decrease in the concentration of sirtuin 1 in the plasma of the studied patients, where triglycerides and CRP were determined as the independent variable for the level of SIRT1. Along with the weight loss, the concentration of SIRT1 in patients decreased, which indicates a significant reduction of inflammation and oxidative stress [117]. Savastano et al. (2015) also pointed to the relationship of SIRT4 with obesity. The reduced level in the blood serum in patients suggests that increasing its activity through the diet may positively reduce the accumulation of adipose tissue [118]. It is assumed that SIRT4 and SIRT1 control insulin secretion in opposite directions. Therefore, research on their effect on type 1 and type 2 diabetes is continued [119]. Introducing a specialized diet with caloric restriction and exercise training as a therapeutic measure requires a very individual approach. Both diet and sport activate hormesis. The dose–response relationship for hormetines is not linear, and individual variability has been found in the Nrf2 nuclear pathway in human studies. Individual variability can be observed in response to dietary hormetins (bioactive phytochemicals, peptides, polysaccharides) and exercise of varying intensity [120] For example, reactive oxygen species, reactive nitrogen species, and reactive lipid species derived from polyunsaturated fatty acids (PUFA) are formed during low-intensity training [121]. There are also interactions between enzymes involved in cytokine regulation. Increased production of antioxidants can cause reductive stress, and some antioxidants have a pro-oxidative effect, reducing the response to physical effort. According to the “mitohormesis” hypothesis, reactive oxygen species (ROS) should be at low levels to produce beneficial effects as signaling molecules that promote viability. Caloric restriction and physical effort become a stress for the cell, which, well-balanced, will cause an appropriate increase in ROS and activation of Nrf2, improving the redox balance [103,122]. To obtain the desired effects, factors such as gender, lifestyle, and general condition of the individual must be considered [107].

### 3.2. FOXO Proteins and P53 Protein

Forkhead box proteins present in mitochondria may interact with sirtuins and affect the processes responsible for aging [123]. The proteins belonging to the FoxO family (forkhead boxO) are transcription factors that play a significant role in the organism at many levels. These include four molecules: FOXO1 (FKHR), FOXO3A (FKHRL1), FOXO4 (AFX1) and FOXO6. Invertebrates have one FOXO gene, while mammals have four: FOXO1, FOXO3, FOXO4, and FOXO6 [124]. Since they are expressed in almost every cell of the body, and many genes encoding FOX proteins have been identified, their biological significance is constantly being studied and supplemented with new information. A characteristic feature of FOX proteins is a specific sequence of 80 to 100 amino acids forming a forkhead box motif, which can bind to DNA [125].

FoxO proteins regulate cell profiling and differentiation and promote programmed cell death [126]. They are involved in response to oxidative stress and are engaged in glucose and lipid metabolism. The activity of FoxO proteins is regulated by post-translational modifications such as phosphorylation, acetylation, and glycosylation. Therefore, changes in the affinity of FOXO to DNA and modification of the transcriptional activity of the FoxO function are involved not only in the pathogenesis of civilization diseases (diabetes, cancer) but can also contribute to the life-extending of organisms by activating longevity genes responsible for the antioxidant’s synthesis or HSP chaperones [127,128,129]. Longevity-regulating genes include the Forkhead transcription factor FOXO and the NAD-dependent histone deacetylase, silent information regulator 2 (Sir2). Thus, sirtuins may regulate the activity of FOXO factors through deacetylation and involvement in DNA damage repair. Studies have shown that, in a sirtuin-dependent manner, a chronic stressor can prolong the life of *S. cerevisiae* and *C. elegans*, and deacetylation of the transcription factor FOXO1a by sirtuin 1 increases the transcription of gluconeogenesis genes and activates transcription factors responsible for the action of insulin genes [130,131]. SIRT1 also reverses H2O2-induced FOXO4 acetylation by inducing the protein GADD45 (DNA damage-induced growth arrest and α) [132], which is involved in cell cycle arrest and DNA repair [133]. In addition, it has been proven that Sir2 can increase the lifespan of individuals by inhibiting homologous recombination of ribosomal DNA [1,126]. Oxidative stress, which reduces the efficiency of cellular repair processes, is managed by deacetylation of FOXO3a via SIRT2 [134], SIRT1 and proper regulation of FOXO1, FOXO3, FOXO4, and p53 proteins [131]. Deacetylation of the p53 protein reduces p53-induced apoptosis, and increased p53 expression is responsible for the symptoms of premature aging in mice [135]. Therefore, the interaction of SIRT1, FOXO, and p53 seems to be of interest in the organism’s survival. Mammalian SIRT1 may bind to FOXO4 and increase its activity by catalyzing acetylation in a NAD-dependent manner. In response to oxidative stress, FOXO accumulates in the nucleus and induces the expression of appropriate genes, such as MnSOD or SOD2. FOXO4 activity can be enhanced by resveratrol or suppressed by a SIRT1 inhibitor, such as nicotinamide. Mechanisms of interaction between sirtuins and forkhead box proteins are complicated. Thus, they are constantly studied in various experimental models [132,136,137,138,139]. Krishnamoorthy and Vilwanathan (2020) have linked SIRT6 to tumor suppression. Researchers believe that the p53/p21 complex mediates the inhibition of lung cancer in humans with simultaneous suppression of sirtuin 6, indicating the need for further research [140]. The p53 protein has tumor suppressor features and can promote aging and cell death [141]. It regulates antioxidant processes by promoting oxidative enzymes and may be associated with cellular metabolism [142]. Kim et al. (2022) showed a link between SIRT1 and the promotion of the p53/p21 complex by lowering acetylation at lysine residues in the C-terminal lysine residues of the p53 protein [141]. Igase et al. (2020) also proved that SIRT1 could inhibit its tumor suppressor functions [143] and affect the production of cytokines in an organism [144].

## 4. Modulators of Sirtuin Activity

Due to the participation of sirtuins in the aging process, determining which compounds have the most effective effect on the modulation of their activity is an urgent issue [145]. The mutually antagonistic mechanisms of sirtuins initiated by proper modulators can prevent many different diseases and thus become a valuable therapeutic agent. Information on sirtuin activators and inhibitors is still unclear, which has changed and even become mutually exclusive over the years. The action mechanism of these compounds is vague, and the different strengths directed against other members of the sirtuin family show that they do not target only one conserved enzyme [146], making their classification difficult. For example, suramin, initially considered a synthetic inhibitor of SIRT1, has also shown inhibition of SIRT5 through virtual screening data. Due to its lack of specificity and chemical multifunctionality, it is not considered for further research [146,147]. Sirtuins engage in many metabolic pathways, so changing their activity may affect one area positively but another negatively. For example, inhibiting sirtuin activity by sirtinol causes platelet aggregation and causes thrombocytopenia, but at the same time may be beneficial in cancer therapy [148]. Sirtinol, similarly to suramin, acts non-specifically by inhibiting SIRT2 and, to a lesser extent, SIRT1 [149].

Studies on the use of sirtuin activators focus on resveratrol (RV), which is found as a flavonoid in food and beverages [150,151,152]. Quercetin and fisetin, as natural polyphenols structurally similar to RV, also have many health benefits and the ability to activate SIRT1. Unfortunately, all of them have been shown to have low bioavailability, which is insufficient for clinical use. Moreover, their metabolites may be pharmacologically active [76,153,154]. In recent years, (until 2022), there has been a noticeable increase in interest in synthetic activators and inhibitors of sirtuins that can support natural modulators in action. Several effective synthetic SIRT1 activators have been developed that mimic the health benefits of resveratrol (SRT1720, SRT1460, SRT2183) and are in clinical trials in combination with other drugs [153].

Gozelle et al. (2022) used compounds based on 5-benzyl-1,3,4-thiadiazole-2-carboxamide, obtaining effective sirtuin 2 inhibitors-ST29 and ST30, which may also potentially support the treatment of cancer [154]. Additionally, Laaroussi et al. (2022) indicated indole analogs EX-527 as cytotoxic to cancer cells and many times more efficient than nicotinamide and other mentioned inhibitors [146,148,155]. Table 1 summarizes the latest sirtuin modulators with effects on selected model species.

### 4.1. Resveratrol—An Antidote for Aging?

**Resveratrol (RSV)** (3,5,4-trihydroksystylben) (Table 2) is a natural chemical compound of plant origin (phytoalexin). It is most found in the skin and seeds of *Vitis vinifera* grapes. It occurs in the form of two geometric isomers -cis and -trans, where only the trans isomer shows biological activity. In red wine, the concentration of resveratrol ranges from 0.1 to 15 mg/L and is 3–10 times higher than in white wine [186]. Resveratrol has a broad, positive physiological effect, but it remains in cells for a limited time. Therefore, its effect may not be sufficient [187]. Due to its nephroprotective effect in diabetes patients, it may reduce the risk of its occurrence. RV is toxic when an inappropriate therapeutic dose is used [188]. It is noted that resveratrol may inhibit L1-RTP, which in turn is dependent on SIRT1, SIRT6, and SIRT7. An increase in SIRT6 was observed after the use of resveratrol as a therapeutic agent [189]. RV in mice increases LSK ex vivo and reduces T cell proliferation in vivo and ex vivo [18]. In addition, its neuroprotective effect is constantly being assessed. Recent research shows that it mediates multiple molecular pathways related to aging and central nervous system function. It contributes to beneficial epigenetic changes that last for generations. It prevents cognitive impairment and cell neurodegeneration primarily due to its potent antioxidant properties [190]. In addition to scavenging free radicals, it modulates synergistic pathways responsible for anti-inflammatory and anti-apoptotic effects [191]. Demonstrated reduced neuroinflammatory stress with decreased pro-inflammatory cytokines (TNF-α) in rats. Moreover, after experimental central nervous system injury, RSV can ameliorate functional deficits when administered on an ad hoc basis [192]. Studies also indicate the inhibitory effect of resveratrol on the activity of SIRT1, SIRT3, and SIRT5, depending on the substrate used [37]. Chou et al. (2022) reported that treatment with resveratrol as a SIRT1 activator might attenuate kidney aging and inhibit oxidative stress [193]. In mice with traumatic brain injury, it increased SIRT1 expression, which activated p38 MAPK phosphorylation and improved their neurological function due to neuronal loss [194]. Yang et al. (2021) noted that resveratrol in mice increased the expression of SIRT1 and induced DNA repair by regulating NBS1 [195].

### 4.2. Polydatin—An Improved Resveratrol?

**Polydatin (PD)** (Table 2), the glycosidic form of resveratrol isolated from *Polygonum cuspidatum* is considered more effective because its concentration in plants can be up to 15 times higher than resveratrol. It is found in wine, grapes, and peanuts [84,196] in two forms, cis- and trans-. Trans-polydatin is attractive due to its anti-inflammatory and antioxidant properties [197]. It has a protective effect against numerous diseases. Like RV, it is promising in treating many diseases of the nervous system due to its neuroprotective effects (Alzheimer’s disease, Parkinson’s disease, brain, and spinal cord injuries, strokes). The polydatin mechanism works through several pathways Nrf2/Keap1/ARE, PI3K/Akt, ERK/MAPK, TLR/NF-κB/TNF-α/ILs or Bax/Bcl-2/caspases Oral administration of polydatin reduces malondialdehyde (MDA) production and increases antioxidant activity (SOD, CAT) [198]. Tong et al. 2020 showed an effective reduction in cognitive impairment caused by chemotherapy. Doxorubicin-induced stress was reduced by activating the NF-κB pathway and reducing apoptosis [199]. Bao et al. (2022) recognized polydatin as an effective agent in liver damage by NF-κB signaling [200]. Jin et al. (2022) showed that polydatin could increase body weight in mice while reducing insulin resistance [201]. It is considered a potential therapeutic agent in case of oxidative stress and radiation-induced lung damage (SIRT 3 activation) [202,203]. Zhang et al. (2017), based on their research and associations with SIRT3 expression, suggested its use as a drug against diabetic cardiomyopathy [204]. Moreover, by upregulating SIRT1 expression, PD may prevent mitochondrial dysfunction and stimulate IRT mRNA. Unfortunately, polydatin show pharmacological disadvantages such as low solubility, low plasma concentration, or rapid chemical degradation, which makes therapeutic applications difficult [205].

### 4.3. Honokiol—An Effective Antioxidant

**Honokiol (HKL)** (Table 2) is a polyphenol extracted from the bark and leaves of magnolia (*Magnolia officinalis*). Classified as an antioxidant and is traditionally used in treating inflammatory diseases [206]. Recent studies indicate its effectiveness against SARS-CoV-2 by inhibiting furin activity and blocking the function of S-protein (spike) [207]. Due to the ability to cross the blood–brain barrier and the blood-cerebrospinal fluid barrier is characterized by significant bioavailability. Studies on a mouse model by Zhou et al. (2022) report its effectiveness in the treatment of neurodegenerative diseases and the ability to improve mitochondrial function and antioxidant properties. It exerts therapeutic effects in several neurological diseases, including amyotrophic lateral sclerosis (ALS), in both in vitro and in vivo models. It improves the viability of NSC-34 motoneuron cells, characterized by mutated G93A SOD1 proteins, and the morphology of mitochondria in SOD1-G93A cells. This effect has also been shown in the spinal cord. Moreover, honokiol extended the lifespan of SOD1-G93A transgenic mice and improved their motor function [208]. As a GABA modulator and CB1 agonist, it is being considered for treating mood disorders. In the treatment of depression and anxiety disorders, it has a selective anxiolytic effect [209]. By SIRT1 expression, HKL modulates endoplasmic reticulum stress and promotes cell viability [208,210]. HKL applied to lung cancer cells destabilizes Hif-1a, induces apoptosis and arrests the G1 phase. Its anticancer properties are regulated by the SIRT3/Hif-1a pathway [211]. Activating SIRT3 alleviates cardiac cell impairment [212]. Li et al. (2016) showed its effectiveness in increasing the activity of SIRT3 in the smooth muscles of the mouse liver and reducing its steatosis [213]. Moreover, the use of honokiol in older mice stopped bone loss, and the effectiveness was dose-dependent [61].

### 4.4. Triclosan-Synthetic Sirtuin Activator

**Triclosan (TCS)** (5-chloro-2-(2,4-dichlorophenoxy)phenol (Table 2) is a synthetic chemical compound with broad bacteriostatic (in higher concentrations also bactericidal), antifungal, and antiviral properties. Due to its properties, it is often used in cosmetics, and it was overused during the COVID-19 pandemic, which raised concerns about environmental safety because it shows high bioavailability through the skin and oral administration [214,215]. Excessive exposure to triclosan may contribute to thyroid homeostasis disorders, metabolic disorders, cardiotoxicity, and increased cancer risk [215,216,217]. It has been proven that triclosan in rodents can impair the immune response and contribute to liver diseases. At environmental concentrations, it modulates the transcriptional response of Nrf2, SIRT1, and SIRT2 in the livers of *Gambusia affinis* fish, negatively affecting their antioxidant system [218]. Szychowski et al. (2022) showed that short and long triclosan exposure increases the expression of SIRT1 and SIRT3 in neurons of the neural cortex of mice, which initially increased the level of neurosteroids in the cells, and then, despite further stimulation of the aryl hydrocarbon receptor (Ahr), caused their significant decrease [217,218]. So far, there is little information in the literature about triclosan and its effect on sirtuins, which requires further research to explain its action mechanism.

### 4.5. Cambinol–SIRT 1 and SIRT 2 Inhibitor

**Cambinol** ((5-((2-hydroxynaphthalen-1-yl)methyl)-6-phenyl-2-thioxo-2,3-dihydropyrimidin-4(1H)-one) (Table 2) is a β-naphthol derivative with an anticancer effect, defined as a sirtuin 1 and 2 inhibitor [219,220] because it inhibits NAD-dependant deacetylases to reduce cell survival under stress. It has anti-inflammatory effects and is involved in the immune response and metabolic control [221]. Chowdhury et al. (2020) identified cambinol as cytotoxic against cancer cells in vitro [222]. Its inhibition of sirtuins 1 and 2 have been shown to increase p53, FOXO3a, and Ku70 acetylation; however, its effect’s complete mechanism is unknown [223]. Its inhibitory effect is related to the simultaneous inhibition of histone 4 and NAD+ [224]. There are papers on more effective inhibitors than cambinol and better selectivity for SIRT 1 and 2, but further studies in this area are required [225].

### 4.6. EX-527—A Precursor of Modern Sirtuin Inhibitors

**EX-527** (6-Chloro-2,3,4,9-tetrahydro-1H-Carbazole-1-carboxamide) (Table 2) as an indole compound is a matrix for synthesizing sirtuin inhibitor analogs. It is mainly used in studies of physiological and cancer cells [149,155]. Nikseresht et al. (2019) showed the relationship of EX-527 microinjections with the reduction of cerebral ischemic infarcts and proved that by inhibiting SIRT1, they act as an inhibitor of necroptosis in an animal model [226]. According to Kundu et al. (2020), it protects the organism against the effects of a high-fat diet (diabetic nephropathy), as it contributes to lowering blood glucose levels and reducing SIRT1 activity while increasing SIRT 3 action in the kidneys [227]. In addition, inhibiting SIRT1 also increases renal allograft survival in mice, which may be related to T cells [228]. It inhibits the activity of SIRT1 100-fold more strongly than SIRT2 and SIRT3, which is presumably associated with NAD+ as a colligand in the binding of EX-527 in cells [149]. Kumari et al. (2015) pointed out that it may cause thrombocytopenia as a side effect in research on anticancer therapy, suggesting while sirtuins are involved in platelet aging and the general aging processes [148].

**Table 2 ijms-24-00728-t002:** Characteristics of selected sirtuin modulators.

Modulator	Chemical Formula	Effect	Characteristic	Reference
Resveratrol	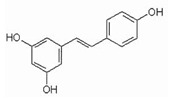	activator	- biological activity in neurogenic diseases; - affects the CNS; - neuroprotective effect; - antiapoptotic effect; - anti-inflammatory effect; - modulates the action of the antioxidant system; - reduces nicotine genotoxicity in *D. melanogaster* by activating sirtuins.	[187,229]
Polydatin	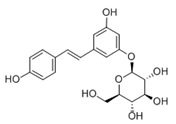	activator	- liver protection and anti-apoptosis; - neuroprotector in chemiotherapy - anti-cancer and anti-inflammatory effects; - inhibition of oxidative stress and removal of ROS; - prevents diseases of the cardiovascular and nervous systems.	[84,199,204]
Honokiol	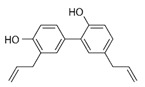	activator	- regulates the process of acetylation of mitochondrial proteins through SIRT; - therapeutic agent against obesity by activating SIRT3 in cells.	[206,208,213]
Triclosan	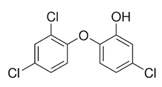	activator	- antiviral and bacteriostatic effect; - widely used in industrial products; - toxic to the environment; - induces the expression of SIRT1 and SIRT3 in mouse neuronal cells; - modulates the expression of SIRT1 and SIRT2 in Gambusia affinis liver.	[230,231,232]
Cambinol	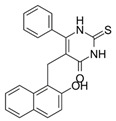	inhibitor	- inhibition of SIRT1 and 2 activity; - inhibition of immune responses and inflammatory reactions; - reduces the activity of NF-κB, which is indirectly dependent on sirtuins.	[219,220,221,233]
EX-527	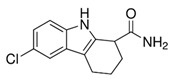	inhibitor	- inhibitor of SIRT 1 and 2, cytotoxic effect in tumor cells; - protective effect in diabetic nephropathy; - reduces blood sugar in a rat model; - inhibitor of necroptosis in animal model (by inhibiting SIRT1)	[155,226,227]

## 5. Diseases and Sirtuin-Based Therapies

The multidirectional functions of sirtuins on biological processes interest the scientific community and the pharmaceutical industry [187]. Because sirtuins play a key role in life processes, they are considered potential participants in modern therapies. Apart from prolonging life, they can also “extend the period of healthy life” (Table 3) [234]. Therefore, the sirtuin diet, “SIRTfood”, based on caloric restrictions, promoting sports, and consuming products rich in sirtuin activators, e.g., resveratrol, is gaining popularity [235]. Abnormal lipid, sugar, or protein economy functioning are common metabolic disorders. Sirtuins affect the maintenance of proper metabolism and may promote the condition of the heart. Cao et al. (2022) point to SIRT3 agonists as promising therapeutics against cardiovascular diseases [63]. In addition, SIRT 1, 2, 6, and 7 reduce myocardial hypertrophy, and SIRT7 may prevent cardiomyopathy [236]. Zaganjor et al. (2021) researched sirtuin 4 and its link to branched-chain amino acid (BCAA) catabolism to treat abnormal amino acid metabolism. Its reduced expression in the adipose tissue of mice has been shown, which is probably a consequence of its ability to increase BCAA catabolism and promote the activity of PPARγ, which regulates glucose metabolism and fatty acid storage in the body [237]. It has been proven that the cellular metabolism of sirtuins is correlated with, e.g., the pathophysiology of bone and joint diseases [238]. Estrogen receptor (ER) deacetylation by SIRT6 in proosteoclasts prevents bone loss and shows promise for the treatment of osteoporosis in elderly or postmenopausal patients [239]. In addition, NAD+, as a metabolite consumed by sirtuins, plays an essential role in pain regulation and peripheral neuropathic pain [240]. He et al. (2022) demonstrated that nuclear and mitochondrial sirtuins might have a neuroprotective/regulatory character in Parkinson’s (PD), Alzheimer’s (AD), and Huntington’s disease [12]. Govindarajulu et al. (2021) showed that AD is associated with aberrant SIRT3 expression [241]. SIRT1 was also shown to be attenuated in the cerebral cortex of AD patients with severe cognitive impairment. The phenomenon is typically associated with the accumulation of amyloid-β and father tau [242]. In addition, it has been observed with signs of low SIRT6 growth, which is related to telomere destabilization. AD is considered the leading cause of senile dementia worldwide, and additional research may contribute to therapeutic benefits [243]. In neurodegenerative processes, apart from oxidative stress, enzymes of histone deacetylase (HDAC), α-synuclein, and autophagy are involved. Parkinson’s disease progresses with age, and the proper function and biological activity of mitochondrial proteins are attributed to SIRT3, SIRT4, and SIRT5 (mitochondrial sirtuins). Mitochondrial sirtuin activators positively affect the stability of the respiratory chain and oxidative stress in neurons. However, clinical trials are insufficient to resolve their complex function in favor of future PD therapies [12,244]. Interestingly, SIRT1 inhibition also gives good results in treating HIV infection or fragile X syndrome [245,246]. As mentioned, sirtuins are involved in metabolic control and epigenetic modification [25]. Moreover, they are significantly associated with diabetic neuropathy and cardiomyopathy [28,247]. Research by Hu et al. (2021) indicated that SIRT1 is a significant regulator responsible for body carbohydrate metabolism disorders [248]. Sirtuins are also responsible for women’s gynecological health because they regulate ovarian function on many levels. They are responsible for the secretion of steroid hormones and maintain normal reproductive functions [249]. Overexpression of SIRT7 with simultaneous repression of SIRT1, SIRT2, SIRT4, and SIRT5 promotes uterine tumor growth [250]. On the other hand, SIRT7 deficiency in the cytoplasm of oocytes and granulosa cells contributes to the inhibition of estrogen synthesis, cell proliferation, and reproductive function [251]. SIRT2 is also responsible for steroid hormone homeostasis and follicle maturation by regulating estrogen and testosterone secretion with its specific inhibitors [252]. In addition, Zhang et al. (2022) suggested a link between SIRT3 activity and polycystic ovary syndrome, as it may regulate glucose metabolism, which plays an essential role in signaling granulosa cells and oocytes. SIRT3 deficiency causes defects in the cellular insulin signaling pathway and contributes to impaired oocyte production [253]. Some sirtuin inhibitors (SIRT1), such as nicotinamide, have the potential to be used in cancer therapy. Sirtuin inhibition leads to an increase in p53-dependent apoptosis and a reduction in tumor cell proliferation [254]. Studies also show the involvement of sirtuins in cellular processes responsible for inflammatory responses, indicating that SIRT1 may modify immune responses in reaction to microbes, reducing the production of pro-inflammatory cytokines in human cells [255,256].

## 6. Sirtuins as an Elixir of Youth?

Aging irreversibly stops somatic cell division and deteriorates viability [270]. It can be a process of various extents and manifests itself in neurodegradation, bone decalcification, chronic inflammatory response, DNA instability, and cancer [271]. With age, various irreversible changes occur in an organism, which may significantly weaken the immune system [18]. The search for an effective “elixir of youth” has been going on since antiquity and attempts to understand the complex mechanisms of aging continue. Sirtuins are products of the Sir-2 gene, which along with the p66shc, ink4a, FOXO, and daf-2 genes, is referred to as the longevity gene [272]. Sirtuins have dual metabolic activity (deacetylation and ADP-ribosylation) coupled with NAD+ hydrolysis. As mentioned earlier, sirtuins regulate many biological processes due to their high substrate specificity and impact on numerous factors. Sirtuins have been shown to play a role in DNA repair, transcription processes, post-translational modification of proteins, maintaining chromosome stability, cell proliferation, apoptosis and cell cycle regulation, control of metabolic processes, gene silencing, and many others. Therefore, they are responsible for the hormonal and metabolic balance and proper work of the nervous, muscular, cardiovascular, digestive, and immune systems [19,273]. They have the potential to counteract the aging process by modulating the response of cells to stress, changing their pattern, and restoring tissue homeostasis responsible for phenotypic aging [6]. Several studies have linked SIRT1, SIRT6, and SIRT7 to life extension in rodents and humans [18,73,274,275]. Most often, studies of sirtuins in terms of longevity concern SIRT1 and SIRT6 due to the correlation of their expression with lifespan in mice, although it is difficult to clearly define which of the sirtuins significantly increases lifespan in vertebrates [276]. Sirtuin 6 has also been evaluated to prevent dendritic cell immune aging [277]. Sirtuins might potentially protect the kidneys, maintaining homeostasis of podocytes damaged with age. However, this theory has not been fully confirmed [17]. Mitochondrial sirtuins may play an essential role in preventing aging, especially in the context of the “mitochondrial theory of aging” and oxidative stress [278]. An indirect anti-aging agent is ergothioneine, which through its antioxidant properties, eliminates mitochondrial oxidative stress, increases the expression of SIRT1 and SIRT6, and reduces hyperglycemia [279]. Reprogramming cells towards self-rejuvenation seems to solve the mystery of longevity and a sought-after element of the philosopher’s stone. However, theories regarding the usefulness of sirtuins as longevity enzymes are not unambiguous. Studies also point to the potential dangers of manipulating sirtuin activity via modulating compounds (e.g., STACs) [280]. Excessive, uncontrolled expression of sirtuins can lead to dysregulation of an organism, promote epithelial-mesenchymal transition and cancer, and cause therapeutic resistance. SIRT genes are upregulated in many types of cancer, presumably due to the dysregulation of positive and negative feedback loops [40,280].

## 7. The Darkside of Sirtuins

The neuroprotective and life-prolonging function of sirtuins is the main direction of research. Until 2022, a few published studies pointed to the effects of the threat caused by modifying the activity of sirtuins and interfering with their mechanisms of action. Preliminary research often leads to erroneous conclusions. Palmer and Vaccarezza (2020) showed that current research on sirtuins has focused on the benefits rather than the potential side effects [280]. Liberale et al. (2020) demonstrated in studies on transgenic mice that SIRT5 might promote venous thrombosis and silencing its expression may be a potential therapeutic in this disease [281]. It has also been proven that overexpression of SIRT1 may cause memory deficit in mice and has no neuroprotective effect [282].

Moreover, SIRT1 is a crucial stimulator of proliferation and angiogenesis. Therefore, its increased expression in women leads to the development of endometriosis and ovarian cancer [283]. He et al. (2022) also confirmed the stimulating effect of SIRT5 on cell proliferation and breast cancer metastasis. Increased expression of SIRT5 is probably related to aerobic glycolysis and the activity of HK2 and PKM2 enzymes in promoting liver and breast cancer [284]. An imprecise and quick application of sirtuins in therapies can cause long-term and unforeseen consequences. Sirtuins involve numerous mechanisms in multicellular organisms, many of which have not yet been described or are based only on general information. To find safe therapeutic solutions, it is necessary to study the mechanisms of action of sirtuins in many experimental models, seeing a holistic effect on the species’ physiology. The phenomenon of hormesis should also be considered [128,285]. Visioli et al. (2022) noted the strategies scientists use to stop the aging process. They recommend optimal dosing of activator substances before clinical trials on humans, as the results obtained from animal models are often inadequate or difficult to interpret [286].

## 8. Looking into the Future—Are We Ready for Longevity?

The future of sirtuins as longevity enzymes remains unclear and difficult to predict. It seems that the topic of compounds that modulate their effects to prolong life and treat diseases remains in the realm of science fiction. Many studies describe their great therapeutic potential, extinguished by many uncertainties and suggestions for further research. For example, sirtuins may simultaneously act as promoters and suppressors in neoplastic diseases. However, these and other conflicting data make the study of sirtuins even more fascinating. There are undoubtedly more reports about new sirtuin modulators and related mechanisms. We know that sirtuin activators and inhibitors are imperfect and require laboratory “discipline”, but the achievements outline the direction to follow to get the way of longevity. Science has an excellent database that gives hope that sirtuins will be included in medicine (Table 3) and used in future anti-cancer strategies [267], thrombotic disease therapies [287], treatment of nephrogenic diabetes [74] or suppressing inflammatory responses [262]. Perhaps, in some time, the next stage will be clinical trials and the widespread availability of drugs. However, caution should be exercised with as many substances because various processes may be associated with sirtuins and alter known and developed strategies. The desire for a healthy and long life promotes overexposure to sirtuin modulators and may paradoxically lead to homeostasis disorders and long-term, unforeseen consequences. Therefore, it is a priority to understand the mechanisms of action of sirtuins and develop their safe modifications. Many scientists believe these processes are too complex to plan our future with sirtuins today.

## 9. Conclusions

Sir2 proteins are involved in a wide range of biological processes and have potential effects in the treatment of lifestyle diseases. The scientists’ curiosity is stimulated by the connection of sirtuins with the length of life. We are certainly at the beginning of understanding the phenomenon of these proteins. Every year new pieces of information are obtained, but there is still a long way to create the perfect anti-aging panacea. Undoubtedly, focusing on sirtuin modulators and understanding the mechanism of their action is currently the biggest challenge. A big step into the future will be the systematic search for compounds with a similar structure to known modulators. Research in silico as a modern strategy in science gives a chance for a holistic view of data obtained by scientists around the world. We believe molecular modeling tools can select the best candidates with enormous potential. Next, work on increasing the bioavailability and retention time of sirtuin modulators should be considered. In the case of some compounds of high biological importance, satisfactory results are obtained by designing structures of the active substance-nanoparticle type. However, biological aging is a complex process affected by many factors, such as environment, species, and individual characteristics of the population. Therefore, scientists will undoubtedly invest a lot of invention, creativity, and effort before solving the mystery of aging and manipulating this process by selected factors.

## Figures and Tables

**Figure 1 ijms-24-00728-f001:**
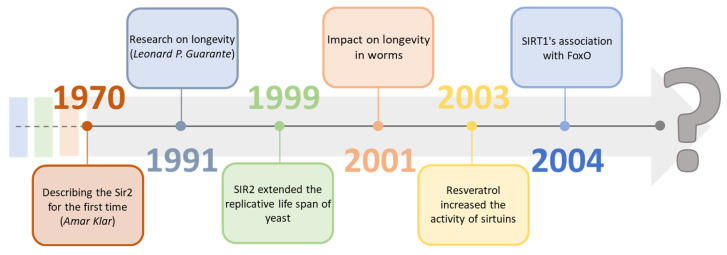
Timeline of sirtuin history [4,5,6,7,8,9,10,11].

**Figure 2 ijms-24-00728-f002:**
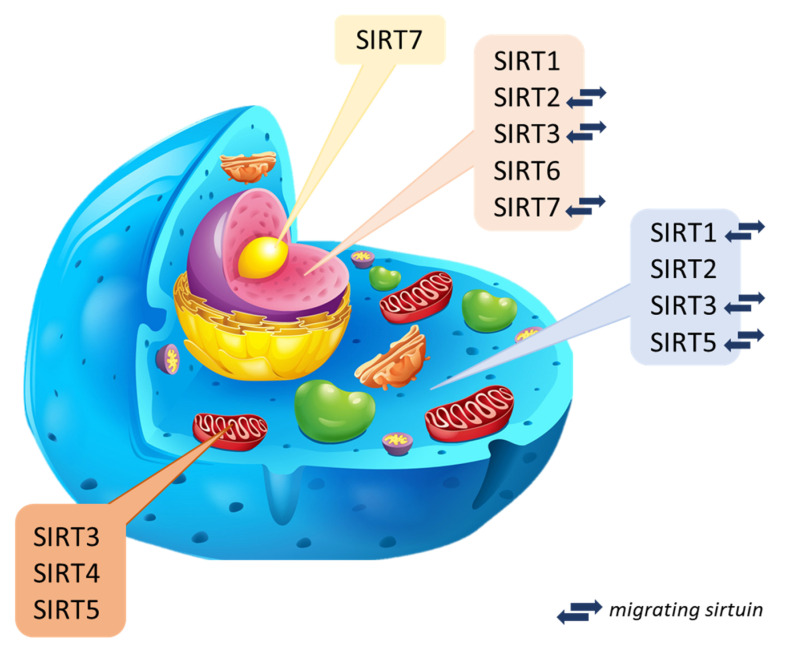
Intracellular localization of mammalian sirtuins [2,34,39,40,41,42,43].

**Figure 3 ijms-24-00728-f003:**
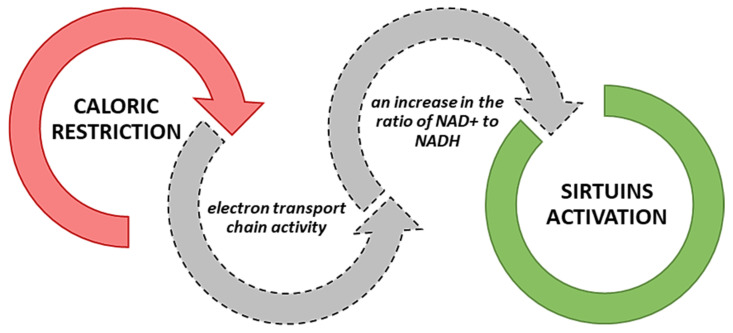
Mechanism of sirtuin activation during caloric restriction [100].

**Table 1 ijms-24-00728-t001:** Summary of studies (from 2017 to 2022) on sirtuin activators and inhibitors (1–7).

Sirtuin	Organism	Modulator	Effect	Reference
ACTIVATORS				
Sirtuin 1	Human and Drosophila melanogaster	ginsenosides (*Panax ginseng* Meyer extract)	- promoting mitochondrial biosynthesis; - modification of the SIRT1 pathway, neuro- and cardioprotective effects;	[156]
	Mammalian cells (*Mus musculus*)	resveratrol	- increased SIRT1 and AMPK activity, improved cardiac function in mice; - reduced expression of parasitic cells;	[157]
		cucurbitacin E glucoside (CuE)	- inhibition of the NF-ĸB/NLRP3 signaling pathway; - enhancing the SIRT1/Nrf2/HO-1 pathway, suppressing oxidative stress in liver cells in mice;	[158]
		quercetin	- SIRT1 increasing activity and PKM2 inhibition;	[159]
	*Rattus norvegicus*	resveratrol	- reducing oxidative stress and reducing the level of cardiac biomarkers; - activation of SIRT1 and protection of heart cells by reducing hyperlipidemia;	[160]
	Humans	1,4-dihydropyridine (DHP)	- activation of SIRT1 in the cells of the nucleus pulposus, protection in osteoarthritis of the spine;	[161]
Sirtuin 2	*Drosophila melanogaster*	food nitrites	- SIRT1 and 2 activators, regulation of metabolism;	[162]
	Human (HepG2 cells)	resveratrol	- SIRT2 activation through the Prx1 factor, antioxidant effect in the aging process;	[163]
Sirtuin 3	*Rattus rattus*	theacrine (1,3,7,9-tetramethyluric acid)	- non-toxic effect, reduction of liver fibrosis and fatty liver in rats by SIRT3 activating;	[164]
	*Mus musculus*	dexmedetomidine	- upregulation of SIRT3, reduction of mitochondrial damage caused by ischemic kidney injury;	[165]
	Human	oroxylin A	- cardioprotective effect by increasing the activity of SIRT3 in myocardial cells while preventing mitochondrial dysfunction in cardiac myocytes;	[166]
		*Retama monosperma* (L.) Boiss extract	- upregulation of SIRT3 in human HaCaT cells, treatment of skin scarring.	[167]
Sirtuin 4	*Mus musculus*	resveratrol	- upregulation of SIRT4 in cells (prolonged exposure), protective effect against excitotoxicity.	[168]
Sirtuin 5	*Mus musculus*	dioscin	- downregulation Of Mir-145-5p expression with upregulation of SIRT5 activity; - reduction of oxidative stress;	[169]
		resveratrol	- recovery of cellular metabolism after subarachnoid hemorrhage.	[170]
Sirtuin 6	*Mus musculus*	glicyryzine	- anti-aging effect, upregulating SIRT6 in 24-month-old mice with cranial osteolysis;	[171]
		UBCS039	- SIRT6 overexpression, treatment of lung injury in mice by promoting M2 bone marrow-derived macrophage;	[172]
		Gami-Yukmijihwang-Tang extract	- regulation of SIRT6 pathways, prevention of histone H3 lysine 56 acetylations.	[173]
Sirtuin 7	*Bombyx mori*	resveratrol	- upregulation of SIRT7; - activation of the SIRT7-FoxO-GST pathway;	[174]
**INHIBITORS**				
Sirtuin 1	Sprague Dawley rats	EX-527	- EX-527 mediates in the p53-SIRT1 pathway, regulation of germ cell apoptosis Cr(VI)-induced;	[43]
	*Rattus rattus*		- decrease of sirt1 activity, memory impairment in rats, oxidative stress as the response to prior quercetin stimulation;	[175]
			- SIRT1 may be linked to morphine addiction in a rat model, and its inhibition could be a potential therapeutic strategy;	[176]
	Bone marrow cells-*Mus musculus*	sirtinol	- inhibiting the activity of sirtuin 1 enhances the effects of astaxanthin, antioxidant and anti-inflammatory effects;	[177]
	*Anser domesticus*	nicotinamide	- inhibition of SIRT1 activity, increased cell proliferation in primary hepatocytes, and involvement in the FoxO1 process.	[178]
Sirtuin 1&2	hepatocellular carcinoma cells	EX-527 & cambiol	- SIRT1 and 2 inhibitors increased the effectiveness of sorafenib treatment targeting HCC cells; - reduction of MRP3 and BCRP expression;	[179]
	colorectal carcinoma HCT 116 (CCL-247) cell line	BZD9L1	- SIRT1 nad SIRT2 inhibitor, anti-cancer activity through the p53 pathway;	[180]
Sirtuin 1–3	murine Neuroblastoma cell line	arekoline (*Areca catechu* extract)	- higher ability to inhibit the activity of sirtuins 1–3 than nicotinamide;	[181]
Sirtuin 2	MCF-7 human breast cancer cell line	thiadiazole derivatives	- new potential SIRT2 inhibitors marked as ST29 and ST30. ST29 showed antiproliferative activity, and ST30 was cytotoxic;	[154]
Sirtuin 3	mass spectrometry of SIRT3	ampicillin trihydrate	- SIRT3 inhibitor, no additional information;	[37]
Sirtuin 4	Human	nicotinamide	- SIRT4 inhibition by homologous modeling and docking of NAD+;	[182]
Sirtuin 5	fluorogenic peptide substrates and human sirtuins	TW-37	- slight inhibition of SIRT5 activity;	[183]
	molecular docking	4-((4-(4-acetamidophenyl)thiazol-2-yl)amino)-2-hydroxybenzoic acid	- strong inhibition of SIRT5 activity, enhanced stability of inhibitors;	[184]
Sirtuin 6	articular chondrocytes from human	EX-527	- downregulation of SIRT6 with oxidative stress formation;	[185]

**Table 3 ijms-24-00728-t003:** Characteristics of selected diseases associated with sirtuins.

Sirtuin	Disease	Model	Effect	Reference
Sirtuin 1	Non-alcoholic fatty liver disease (NAFLD)	C57BL/6 mice	- bariatric surgery with sleeve gastrectomy in mice contributed to upregulating the NRK1/NAD+/SIRT1 pathway, alleviating NAFLD;	[257]
	Coronary heart disease (CHD)	Rat cell lines	- inhibition of miR-323-3p or overexpression of SIRT1 resulted in a reduction of cardiac muscle fiber disorder with simultaneous inhibition of vascular endothelial cell apoptosis;	[258]
	Traumatic brain injury (TBI)	Male rats	- increasing SIRT1 expression by regulating PGC-1α expression (upregulated in TBI) may reduce neuronal damage and inhibit microglial activity;	[259]
	Parkinson’s disease	Male C57BL/6JNarl mice	- activation of SIRT1 with SRT1720, reduced cytotoxicity in PD with simultaneous down-regulation of PGC-1α level;	[260]
Sirtuin 2	Non-alcoholic fatty liver disease (NAFLD)	Mice	- reducing the amount of SIRT2 in the liver may contribute to hepatocyte damage. It decreased the expression of GLUT2, FATP2 and FAPT5;	[52]
	Psoriasis	Mice and Human embryonic kidney 93T cell line	- reduced amount of SIRT2 contributes to the pathogenesis of psoriasis development through the effect of sirtuin on the function of Th17 cells, which trigger an abnormal proliferation of KCs;	[261]
Sirtuin 3	inflammatory responses elicited by lipopolysaccharide	Bovine mammary epithelial cells	- presumably, increased expression of SIRT3 is involved in the regulation and amelioration of inflammatory responses as it affects oxidative stress while suppressing the PGC1α-NFκB pathway responsible for the inflammatory response;	[262]
	Chronic Constriction Injury–Induced Neuropathic Pain	Mice	- SIRT3 overexpression resulted in neuropathic pain relief by inhibiting CaMKIIα phosphorylation;	[263]
	Polycystic ovary syndrome (PCOS)	Sprague-Dawley rats	- increased expression of SIRT3 because of the use of Bu-Shen-Tian-Jing Formula (BSTJF) reduced the pathogenesis of polycystic ovary syndrome with concurrent regulation of p38 MAPK and PI3K/AKT signaling pathways;	[264]
Sirtuin 4	Doxorubicin-Induced Cardiotoxicity (DIC)	Male C57BL/6 mice	- SIRT4 overexpression activated the Akt/mTOR signaling pathway, thanks to which it alleviated DIC with simultaneous improvement of cardiac function;	[68]
	Diabetic nephropathy		- SIRT4 activated by FOXM1, inhibits the NFκB pathway alleviated kidney damage.	[265]
Sirtuin 5	hepatic ischemia and reperfusion injury	Male C57BL/6J mice	- increasing the level of SIRT5 causes attenuation of liver damage through the mechanism of inhibition of oxidative stress, raising the level of IHH2 and SOD1;	[266]
	Pancreatic Cancer	PDAC cell lines, genetically engineered mice models, and human tissue samples	- activation of SIRT5 with MC3138, showed anticancer effects in human cells by regulating glutamine/glutathione metabolism.	[267]
Sirtuin 6	type 1 diabetes	C57BL/6 mice	- SIRT6 regulates FoxO1 in renal proximal tubules by addects renal glucose reabsorption and gluconeogenesis in type 1 diabetes;	[74]
	neuroinflammation and ischemic brain injury	Cell culture from C57BL/6 mice and human blood cells	- activation of SIRT6 with MDL-811 has been shown to have neuroprotective and ameliorating effects on neuroinflammation, which is linked to the SIRT6/EZH2/FOXC1 signaling pathway.	[268]
Sirtuin 7	Chronic kidney disease (CKD) with hypertensive	Mice	- in hypertensive SIRT7 alleviates kidney damage (kidney ferroptosis, pathological fibrogenesis, renal dysfunction) by facilitating the KLF15/Nrf2 signaling;	[82]
	colorectal cancer	Human CRC cell lines HCT-8, HCT-116, DLD1, SW620 and HT15	- increased expression of SIRT7 by GRIM-19, affects the SIRT7/PCAF/MDM2 axis, increasing p53 stabilization.	[269]

## Data Availability

Not applicable.

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
