# Peer review of "Why Is Longevity Still a Scientific Mystery? Sirtuins—Past, Present and Future"

_ijms, 2022, doi:10.3390/ijms24010728_

Round 1

Reviewer 1 Report

This study addresses a very interesting and important issue for understanding the molecular mechanisms underlying aging and associated diseases.

The manuscript is fluent. However, there are a couple of main concerns:

1) "caloric deficit" does not appear as an appropriate term; it should be changed to caloric restriction, including in the figure.

2) Caloric restriction is described as a natural sirtuin activator, in contrast, the other established activator which is exercise training is not discussed (only in a phrase like "sport"!).

This aspect is crucial because the activation of sirtuins in the training regimens is important as a possible therapeutic approach and to substantiate and, at the same time, investigate the phenomenon of hormesis, a concept that needs to be discussed better in the manuscript.

(There are several studies that have addressed these issues: Nutrients. 2022 May 17;14(10):2092. doi: 10.3390/nu14102092. ; Aging (Albany NY). 2020 Apr 22;12(8):6852-6864. doi: 10.18632/aging.103046. J Inflamm Res. 2021 Apr 7;14:1283-1296. doi: 10.2147/JIR.S300997. 

Minor

Some typos and errors need to be corrected by doing an English language refresh.

Reviewer 2 Report

The manuscript entitled Why is longevity still a scientific mystery? Sirtuins – past, present and future is a well-written review. My suggestions to improve the manuscript are;

1. In sections 4.1-4.6 more recent studies connected with ageing and neuroprotective effects may be included, For instance, Kinra, M., Ranadive, N., Mudgal, J., Zhang, Y., Govindula, A., Anoopkumar-Dukie, S., Davey, A.K., Grant, G.D., Nampoothiri, M. and Arora, D., 2022. Putative involvement of sirtuin modulators in LPS-induced sickness behaviour in mice. Metabolic Brain Disease, pp.1-8.

2. Section 5 needs improvement. Diseases and sirtuin-based therapy. It will be good to develop a table explaining the disease, model studied,  sirt involved and mechanism. 

3. Even though the title suggests a future of sirtuins I couldn't find a future perspective of sirtuins in the paper. This is a major drawback. Suggest including therapeutic delivery system/interventions which might be able to target sirtuins to improve longevity.  

4. Figures explaining signalling mechanisms targeted by SIRT are required. 

Round 2

Reviewer 1 Report

The authors have adequately responded to requests
In my opinion, the manuscript is suitable for publication.